# Epidemiology of Keratinocyte Skin Cancer with a Focus on Cutaneous Squamous Cell Carcinoma

**DOI:** 10.3390/cancers16030606

**Published:** 2024-01-31

**Authors:** Lena Nanz, Ulrike Keim, Alexander Katalinic, Thomas Meyer, Claus Garbe, Ulrike Leiter

**Affiliations:** 1Center for Dermatooncology, Department of Dermatology, University of Tübingen, Liebermeisterstr. 25, 72076 Tübingen, Germany; lena.nanz@med.uni-tuebingen.de (L.N.); ulrike.keim@med.uni-tuebingen.de (U.K.); claus.garbe@med.uni-tuebingen.de (C.G.); 2Institute for Social Medicine and Epidemiology, University of Lübeck, Maria-Göppert-Str. 22, 23562 Lübeck, Germany; alexander.katalinic@uksh.de; 3Department of Dermatology, Venerology, and Allergology, University of Bochum, Gudrunstr. 56, 44791 Bochum, Germany; thomas.meyer@klinikum-bochum.de

**Keywords:** incidence, mortality, keratinocyte skin cancer, cutaneous squamous cell carcinoma

## Abstract

**Simple Summary:**

The incidence of keratinocyte skin cancer has rapidly been increasing over the last five decades in fair skinned populations throughout the world. About 20% account for squamous cell carcinoma (SCC), and mainly the elderly are affected. Although the mortality rate is low, keratinocyte skin cancer is associated with a high morbidity, especially if multiple tumors occur, and pose a problem for the healthcare system. Here, we present the epidemiology of keratinocyte skin cancer, with a focus on SCC in Queensland, Australia; the United States and the north of Europe, and give an outlook to further challenges.

**Abstract:**

Keratinocyte skin cancer, consisting of basal cell carcinoma (BCC) and squamous cell carcinoma (SCC), is by far the most common cancer in white-skinned populations, with rapid increases over the last 50 years. While the age-standardized incidence rates increase worldwide, the age-standardized mortality rates are variable. The incidence rates of keratinocyte skin cancer are much higher compared to those of melanoma, and are largely attributed to the raising exposure to ultraviolet (UV) radiation, the most important causal risk factor for skin cancer. Whereas the development of BCC is mainly due to intense UV exposure during childhood and adolescence, the development of SCC is related to chronic, cumulative UV exposure over decades. Although mortality rates are relatively low, SCC is an increasing problem for healthcare services, significantly causing morbidity, especially in older age groups. This review reports on the epidemiology of keratinocyte skin cancer, with a focus on SCC, in Australia, the United States, and the north of Europe, with an outlook on further challenges health systems will be confronted with in the next 20 years.

## 1. Introduction

Nonmelanoma skin cancer is currently the most common malignant neoplasm in white populations and thus in Germany as well [1,2]. On the other hand, it is rare in African and Asian populations because these populations have strong protection by pigments [3].

Nonmelanoma skin cancer mainly consists of keratinocyte skin cancer, with about 80% basal cell carcinoma (BCC) and about 20% squamous cell carcinoma (SCC), which originate from keratinocytes [3,4]. Other nonmelanoma skin cancers, like adenocarcinoma resulting from gland cells of the skin, sarcoma, Merkel cell carcinoma, and other cancers, each account for 0.2 to 0.4% of all nonmelanoma skin cancers [1]. Therefore, if incidence rates of nonmelanoma skin cancers are given, they primarily refer to keratinocyte skin cancer (BCC and SCC).

In white populations, all skin cancers are predominantly caused due to ultraviolet (UV) radiation. This means that they could be largely prevented through behavioral modification and avoidance of UV exposure [3]. Considering the development of keratinocyte skin cancer, the importance of UV radiation also mirrors the mutation patterns of these tumors. Examination of the mutational burden in different tumors revealed the following: hematologic and pediatric tumors showed the lowest mutational burden, while the highest mutational burden was detected in lung cancer and melanoma. These are typically induced by exogenous carcinogens, such as cigarette smoke and UV radiation [3,5]. Further studies showed that the mutation load is significantly higher in SCC than in melanoma [3]. In SCC, 61 mutations per Mb were found, while in melanoma only 13 mutations per Mb were found [6]. SCC of the skin is a malignant neoplasm of the keratinocytes of the epidermis and can develop different degrees of differentiation. The diagnosis has to be confirmed histologically, either by an excisional biopsy or a therapeutic excision. SCC of the skin arises in most cases, but not necessarily, as a result of intraepidermal proliferation of atypical keratinocytes. Invasive SCC is said to occur when there is histomorphologically demonstrable disruption of the basal membrane beneath an intraepithelial keratinocytic proliferation in non-traumatized skin. In contrast, BCC is histologically characterized by cells that are similar to an embryonic hair germ. Basaloid cells have a basophilic oval nucleus with an inconspicuous nucleolus and narrow cytoplasm. The tumor cells are surrounded by a fibromucinous stroma. A palisade-like arrangement of the tumor cells at the edge of the tumor cell aggregates is typical. The tumor cells are in contact with the surface epithelium. Immunohistochemically, BCC reacts with pancytokeratin markers such as MNF116, and is cytokeratin 5/6- and cytokeratin 17-positive. BerEP4 is used for differential diagnosis, and BCC usually stains consistently with the antibody. As a diagnostic tool, dermoscopy is a suitable procedure for differentiating other diseases and tumors in the case of unclear findings. Other non-invasive imaging procedures include confocal laser microscopy and optical coherence tomography [7].

The incidence of nonmelanoma skin cancer is far higher than that of any other skin cancer. In Germany, the incidence of nonmelanoma skin cancer is about tenfold-increased compared to melanoma. For the year 2012, 25 cases of melanoma and 250 cases of nonmelanoma skin cancer per 100,000 person-years were registered [8,9]. However, the mortality rate is very low [1]. For this reason, nonmelanoma skin cancer is usually not recorded by cancer registries worldwide, and others report only the first tumor [4]. For example, there are no cancer registry data on nonmelanoma skin cancer in the United States [3]. In order to obtain data on nonmelanoma skin cancer, health insurance data were analyzed. It turned out that 2,000,000-to-3,000,000 procedures were billed annually for the treatment of nonmelanoma skin cancer [10]. In Germany, nonmelanoma skin cancer is recorded by several cancer registries in different federal states. We refer here to the rates of the federal states of Schleswig Holstein and Saarland. The Schleswig Holstein registry shows the highest incidence of nonmelanoma skin cancer, and the Saarland registry has the longest duration of cancer registration in Germany [9].

In this review, epidemiological data on cutaneous SCC were reported separately, as SCC compared to BCC has a higher risk of recurrence, morbidity, and mortality. Therefore, an analysis of epidemiology and future trends was of interest to assess the current and forthcoming burden on dermatologic healthcare.

Reporting on the incidence and mortality rates, age-standardized incidence rates, and age-standardized mortality rates are indicated per 100,000 person-years. For each region, the rates given here were calculated according to the age-standard population in the respective regions.

## 2. Incidence of Keratinocyte Skin Cancer

In Australia, the incidence of keratinocyte skin cancer is very high, with the most extreme rates in Queensland, as shown in a study from 2011 to 2014 that analyzed Australian Medicare data. Here, the Australian standard population (2001) was used to estimate the age-standardized incidence rates of keratinocyte skin cancer. In Queensland, the age-standardized incidence rate of both sexes together amounted to 2679 per 100,000 person-years. Men had a considerably higher age-standardized incidence rate than women, at 3105 per 100,000 person-years and 2296 per 100,000 person-years, respectively. Further, this study implied that during the study period, 47% of patients had multiple keratinocyte skin cancers, which was highly correlated with age. The age-specific incidence increased from 26 per 100,000 person-years in the age group of 20-to-24 years to more than 6000 per 100,000 person-years in the age group of 80-to-84 years [11].

In the United States, the incidence rates of nonmelanoma skin cancer raised since 1990 according to an analysis that evaluated the Global Burden of Disease 2019 database. The age-standardized incidence rate grew from 402 per 100,000 person-years in 1990 to 787 per 100,000 person-years in 2019 for keratinocyte skin cancer. During this period, males had a significantly higher incidence every year. In 2019, their incidence was 1021 per 100,000 person-years versus females with 603 per 100,000 person-years. The highest age-standardized incidence rates were found in Florida, followed by Arizona and Utah, and were generally higher in the southeast, northeast, and western halves of the United States [12].

Geographic incidence differences were also assessed in the United Kingdom using data from England’s cancer registry and data from Scotland, the north of Ireland, and Wales between 2013 and 2015. The incidence rates of first keratinocyte skin cancer per patient per year were the highest in the southern and coastal areas [13].

Increasing incidence rates of keratinocyte skin cancer have also been announced in other northern European countries. In Denmark, the incidence rates of keratinocyte skin cancer were 126.5 per 100,000 person-years for men and 124.8 per 100,000 person-years for women in 2012 [3]. Although nonmelanoma skin cancer is not routinely collected by cancer registries, in Denmark, they are comprehensively registered in two nationwide population-based registries. Using the combined data from these two registries, it was shown that the incidence of nonmelanoma skin cancer raised from 46.2 to 121.2 per 100,000 person-years between 1978 and 2007, according to the World standard population [14].

Similar results are established for Germany, where the age-standardized incidence rates were given based on the European standard population. In 2014, the age-standardized incidence rates of keratinocyte skin cancer were observed to be 113.2 per 100,000 person-years for males and 85.1 per 100,000 person-years for females [3]. With data from the population-based cancer registry of North Rhine Westphalia, Germany, a raising trend of age-standardized incidence rates for nonmelanoma skin cancer from 2007 to 2015 was demonstrated, with estimated annual percentage changes of 3.6% among men and 5.2% among women [15]. In the German federal state of Saarland, between 1970 and 2016, the age-standardized incidence rates of keratinocyte skin cancer grew from 12.0 to 115.6 per 100,000 person-years in males, and from 9.7 to 102.7 per 100,000 person-years in females. Between these years, age-specific incidence rates increased steadily. The highest incidence rates were observed in persons aged older than or equal to 80 years during this time. Here, the incidence rates increased from 85.3 to 950.1 per 100,000 person-years for men, and from 126.8 to 554.5 per 100,000 person-years for women. Lower incidence rates were noted in persons aged younger than or equal to 40 years. In this group, the incidence raised for both sexes from less than 0.01 to 6.4 per 100,000 person-years in males and to 11.1 per 100,000 person-years in females, respectively [3].

The age-standardized incidence rates of keratinocyte skin cancer in the Australian state of Queensland, the United States, Denmark, and the German federal state of Saarland are listed in Table 1.

## 3. Incidence of Cutaneous Squamous Cell Carcinoma

Cutaneous SCC is usually associated with advanced age, with a mean age at diagnosis of 70 years. More than 80% of cases occur in people aged older than or equal to 60 years. Men are affected about twice as often as women [16,17]. This phenomenon is seen worldwide.

Once more, Australia showed the highest incidence rates of cutaneous SCC, with the greatest rates in Queensland. Here, the age-standardized incidence rates of both sexes together were 467 per 100,000 person-years, with 573 per 100,000 person-years for men and 371 per 100,000 person-years for women from 2011 to 2014 [11].

The incidence rates of cutaneous SCC were lower in the United States. In Minnesota, a study in which the age-standardized incidence rates were calculated adjusted to the population structure of the total United States population (2010) revealed increasing incidence rates with age for females and at a faster rate for males from 2000 to 2010. Age-standardized incidence rates of 207.5 per 100,000 person-years for men and 128.8 per 100,000 person-years for women were observed. In contrast, between 1976 and 1984, the age-standardized incidence rates were presented by 96.2 per 100,000 person-years for males and 35.3 per 100,000 person-years for females [18]. From 2005 to 2019, the incidence rates remained stable, at 262 per 100,000 person-years. In 2019, the United States reported 1.5 million new cases of cutaneous SCC, compared to BCC and melanoma with 2.8 million and 82,054 new cases, respectively [12].

In Europe, lower rates were found in northwestern countries like the United Kingdom. Age-standardized incidence rates, based on the European age standard population of 77 per 100,000 person-years from 2013 to 2015, were reported for cutaneous SCC, showing rates of 111 per 100,000 person-years for men and 42 per 100,000 person-years for women. The highest rates were noted in the north of Ireland. During the 3-year period, the estimated annual percentage change was around 6% [13].

Similar rates were observed in the Netherlands. In an analysis with data from the Netherlands’ cancer registry, the European standard population (2013) was used to estimate the age-standardized incidence rates of cutaneous SCC. The age-standardized incidence rates grew from 35.9 per 100,000 person-years in 1989 to 108 per 100,000 person-years in 2021 for males, while females showed a stronger increase in incidence from 13.1 to 69.7 per 100,000 person-years [19]. Another study from 1989 to 2008 showed that the incidence rates have increased sharply in women since 2002 and in men since 2003, for which there is no simple explanation [20].

In Denmark, the incidence rates of cutaneous SCC were evaluated over a period of 30 years. Between 1978 and 2007, the age-standardized incidence rates increased from 9.7 to 19.1 per 100,000 person-years for males, and from 4.6 to 12.0 per 100,000 person-years for females. This represented an increased estimated average annual percentage change of 3.09% for men. Remarkably, it showed a significantly higher increase for women, at 4.25%, and was supposed to be caused by an increase in tanning beds [14].

In Germany, the Robert Koch Institute estimates that in 2014, approximately 29,300 males and 20,100 females were diagnosed with cutaneous SCC for the first time [3]. For example, in the cancer registry of the federal state of Saarland, data are reported in detail. The age-standardized incidence rates increased from 26.3 to 83.8 per 100,000 person-years in men, and from 14.9 to 35.0 per 100,000 person-years in woman between 1990 and 2019, according to the European standard population (2013). A further analysis confirmed the continuously raising rates from 2001 to 2019. Extrapolations showed that by 2044, the incidence rates are expected to increase to 94.2 per 100,000 person-years in men and 37.8 per 100,000 person-years in women. An analysis according to age groups reported the highest incidence rates among persons aged older than or equal to 60 years. This was especially true for males aged older than or equal to 80 years, where rates increased from 163 per 100,000 person-years in 1990 to 847 per 100,000 person-years in 2019. Considerably a smaller increase, rising from 127 to 285 per 100,000 person-years, was seen under women [19]. Similar age-standardized incidence rates were found in the federal state of Schleswig Holstein, where the age-standardized incidence rates increased from 39.0 to 68.6 per 100,000 person-years in men, and from 19.6 to 36.1 per 100,000 person-years in women between 1999 and 2020, based on the European standard population (2013). For both sexes, a continuous increase in incidence rates can be seen: for women until 2013 and for men until 2015. However, according to extrapolations for the next 20 years, an ongoing increase in age-standardized incidence rates is to be expected up to 115 per 100,000 person-years in males and 63 per 100,000 person-years in females until 2044. Age-specific incidence rates raised for both sexes. Looking at specific age groups, the strongest increases were revealed in people aged older than or equal to 60 years, albeit to a lesser extent from 2015 onward. For men aged older than or equal to 80, the rates doubled from 308 per 100,000 person-years in 1999 to around 600 per 100,000 person-years in 2020, a smaller increase was found in women that grew up from 173 to 278 per 100,000 person-years for the same period [19].

The age-standardized incidence rates of cutaneous SCC in the Australian state of Queensland, the US state of Minnesota, the United Kingdom, the Netherlands, Denmark, and the German federal states of Saarland and Schleswig Holstein are listed in Table 2, and the incidence trends of Saarland and Schleswig Holstein are shown in Figure 1 and Figure 2, respectively.

## 4. Mortality of Keratinocyte Skin Cancer

Compared with incidence, mortality from keratinocyte skin cancer is rather low. Relatively constant estimated age-standardized mortality rates for nonmelanoma skin cancer were described in Australia between 1982 and 2016, according to the Australian standard population (2001). In 2016, the estimated age-standardized mortality rates were 2.8 per 100,000 person-years for men and 1.1 per 100,000 person-years for women. However, age-specific mortality declined for those younger than 80 years of age, apart from those aged from 40 to 49 years, and increased for those older than or equal to 80 years of age from 20 to 34 per 100,000 person-years [21].

Similarly, in the United States, mortality rates from cutaneous SCC stayed unchanged over the past 15 years. For keratinocyte skin cancer, mortality rose from 0.7 to 0.8 per 100,000 person-years between 1990 and 2019. Men had a significantly higher mortality rate every year during this period. In 2019, the mortality rates were 1.3 per 100,000 person-years in males and 0.3 per 100,000 person-years in females. Generally, mortality from keratinocyte skin cancer was higher in the southern half than in the northern half of the United States, with the highest mortality rates in Tennessee, Oklahoma, South Carolina, Kentucky, and Delaware [12].

In contrast to other countries, age-standardized mortality rates in the Netherlands decreased slightly or remained stable over time. They decreased from 1.30 per 100,000 person-years in 1989–1991 to 1.21 per 100,000 person-years in 2018–2020 in males, and were stable in females, at 0.57 per 100,000 person-years [19].

Comparably to the results in Denmark, in Germany, age-standardized mortality rates increased both in the federal state of Saarland and in the federal state of Schleswig Holstein from 1990–1992 to 2017–2019 and from 1999–2001 to 2016–2018, respectively. In Saarland, they increased from 1.30 to 1.79 per 100,000 person-years for men, and from 0.56 to 0.68 per 100,000 person-years for women. This corresponds to an increase greater than 30% in 10 years [19]. In Schleswig Holstein, age-standardized mortality rates raised from 0.96 to 1.60 per 100,000 person-years in males, and from 0.33 to 0.81 per 100,000 person-years in females [19].

The all-cause mortality after BCC or SCC diagnosis was contrasted in a meta-analysis of four countries: Germany, Denmark, the United States, and the Netherlands. For BCC, the mortality rate ranged from 0.87 to 0.97 per 100,000 person-years, so they were quite similar. For SCC, the rates were higher, reaching 1.17 per 100,000 person-years in Germany, 1.30 per 100,000 person-years in Denmark, 1.25 per 100,000 person-years in the United States, and 1.27 per 100,000 person-years in the Netherlands. Thus, compared with the general population, patients with SCC had a 25% increased risk of all-cause mortality [20,22,23,24].

The age-standardized mortality rates of keratinocyte skin cancer in Australia, the United States, the Netherlands, and the German federal states of Saarland and Schleswig Holstein were accessed from national cancer registries, the University of Washington Institute for Health Metrics and Evaluation for the United States, and the Australian Institute of Health and Welfare (AIHW) National Mortality Database of 2013, and are listed in Table 3.

## 5. Challenges in the Registration of Nonmelanoma Skin Cancer

Nonmelanoma skin cancer is usually not recorded by cancer registries worldwide. In the United States, for example, there is a database collecting cancer data, but it does not record nonmelanoma skin cancer. Therefore, data must be requested via health insurance companies, but this is not comparable with the documentation in cancer registries. In the northwest of Europe, there are existing cancer registries, but they have not been recording nonmelanoma skin cancer for a long time [3].

Another problem is that there is a major underestimation of the real skin cancer burden, as many registries only collect the first nonmelanoma skin cancer. A recent study demonstrated that due to changes in registration procedures such as the introduction of an automatic linkage system and especially the inclusion of multiple SCCs, the incidence rates of cutaneous SCC have increased significantly in the Netherlands since 2017. When including multiple cutaneous SCCs per patient, the age-standardized incidence rates have raised by 58.4% for men and 34.8% for women [25].

A further challenge is the recording of tumor-associated deaths. Disease-specific mortality is not reliable because it is documented based on death certificates. As patients are mostly old with significant comorbidities, it is often impossible to determine the cause of death. Therefore, there could be an underestimation of the mortality of nonmelanoma skin cancer [19].

When skin cancer screening was introduced and refunded by health insurers in July 2008 in Germany, higher detection rates and a further increase in observed incidence rates were noticed. As nonmelanoma skin cancer often causes no symptoms or does not lead to death during life, it is frequently not detected until patients participate in screening programs. Early forms of skin cancer, that never damage patients, are especially diagnosed and then treated [19]. Patients older than or equal to 60 years of age are the most likely to participate in skin cancer screening programs compared to other age groups, at around 36% participation [26].

## 6. Expectations for the Future

Although not recorded or presented in many cancer registries, most patients develop multiple cutaneous SCCs in further course. Furthermore, UV-related skin cancer can basically affect anyone exposed to sunlight. The number of skin cancer patients is likely to increase in the future due to demographic developments and climate-related changes in UV exposure, including low ozone events increases on cloudless days, as well as changes in leisure time behavior during more sunny days. Low ozone events, also known as mini ozone holes, can occur over short periods in the Northern Hemisphere when polar air moves south after cold Arctic winters. Since the ozone content over the Arctic is low at this time, air masses with low ozone concentrations can be temporarily shifted to northern and central Europe, leading to sudden high UV radiation in the spring [27,28]. These low ozone events are relevant to health because the skin is particularly sensitive in the spring. High UV exposure is not usually expected at this time of year and sun protection measures are therefore generally not considered. According to the records of the German Weather Service (DWD), the number of cloudless days in Germany has increased in recent years and has led to an increase in the annual sunshine duration. As a result, more UV radiation reaches the surface of the earth and, accordingly, the annual erythema-effective UV irradiance has increased in years with increased sunshine duration [29]. On the other hand, climate change can actually cause more clouds due to extreme weather events at times of the year when UV is the highest. In hot climates such as Queensland, warmer weather is likely to lead to reduced exposure as people retreat indoors to avoid the heat, whereas the reverse may be true in more temperate climates.

Another factor potentially influencing people’s UV exposure is the impact of weather changes on people’s leisure activities. Perceived temperatures in the warm range, as well as cloudless weather mean that many people spend more time outdoors, and thus have a higher UV exposure [30,31,32].

Therefore, an increase in the current and future burden of disease is expected, a challenge that health systems must address. New skin cancer strategies are needed. This means that patients at a high risk for skin cancer and subsequent cutaneous SCC, such as patients with iatrogenic or disease-related immunosuppression, patients with organ transplantation, CLL, or HIV, should be identified, and efficient screening programs with an early detection of new cutaneous SCCs should be implemented for this group in particular [19].

Primary prevention is still inadequate and needs to be improved to prevent further increases in the incidence of cutaneous SCC. There will be no sign of a trend reversal in the foreseeable future. The incidence of cutaneous SCC is expected to increase by 10 to 75% over the next 20 years. Since many people have already been exposed during the past periods of high levels of UV radiation, sufficient to develop skin cancer in the future, a further increase seems plausible. After previous DNA damage to the skin, a latency period of 20-to-30 years is relevant for the development of skin cancer.

## 7. Conclusions

Keratinocyte skin cancer is the most common type of cancer in white populations, with SCC accounting for about 20% of keratinocyte skin cancers. Increasing incidence rates are observed worldwide, which is due to an increase in exposure to UV radiation. For cutaneous SCC, regular and cumulative UV exposure is causative. Mortality rates are low, but their growth is described as uneven. A better coverage of all keratinocyte skin cancer in cancer registries would possibly also increase mortality rates. Nevertheless, keratinocyte skin cancer represents a burden for the healthcare systems, and preventive measures are urgently needed. In western Europe, Australia, and the United States, a further increase in incidence is expected in the coming years.

## Figures and Tables

**Figure 1 cancers-16-00606-f001:**
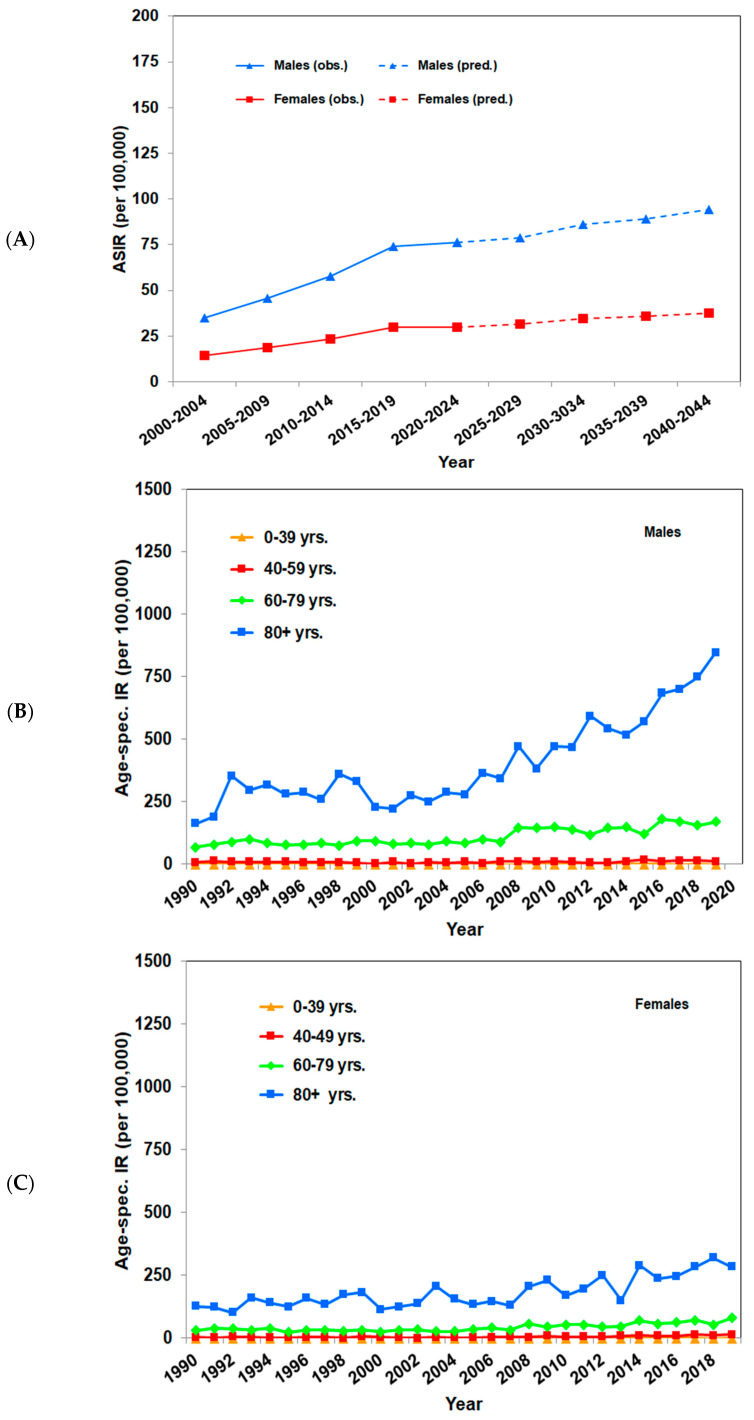
Incidence trends of cutaneous squamous cell carcinoma in the German federal state of Saarland. Figures were calculated according to [19]. (**A**) Observed and predicted age-standardized (EU-27 + EFTA Standard Population, 2013) incidence rates per 100,000 person-years of SCC in intervals from 2000–2004 to 2040–2044. (**B**) Observed age-specific incidence rates per 100,000 person-years of SCC per year for males from 1990 to 2019. (**C**) Observed age-specific incidence rates per 100,000 person-years of SCC per year for females from 1990 to 2019.

**Figure 2 cancers-16-00606-f002:**
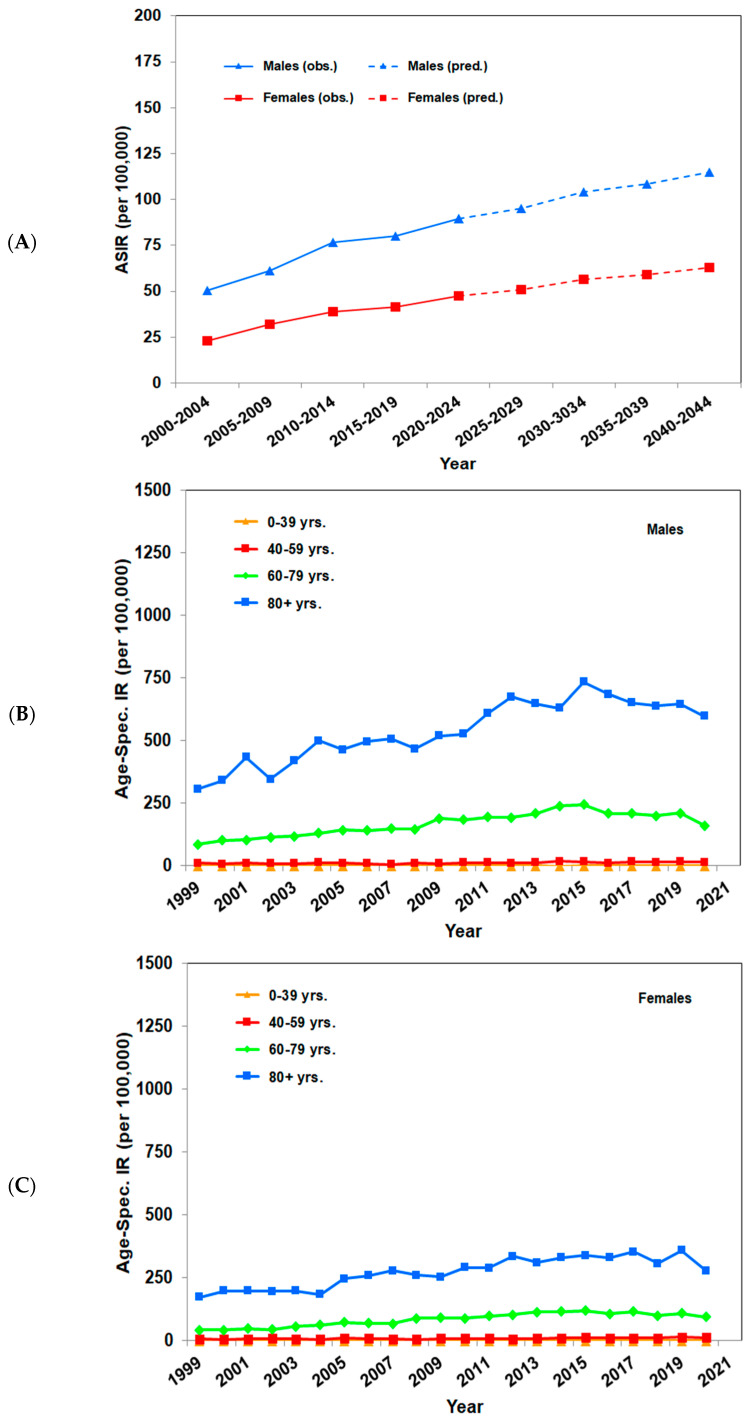
Incidence trends of cutaneous squamous cell carcinoma in the German federal state of Schleswig Holstein. Figures were calculated according to [19]. (**A**) Observed and predicted age- standardized (EU-27 + EFTA Standard Population, 2013) incidence rates per 100,000 person-years of SCC in intervals from 2000–2004 to 2040–2044. (**B**) Observed age-specific incidence rates per 100,000 person-years of SCC per year for males from 1999 to 2020. (**C**) Observed age-specific incidence rates per 100,000 person-years of SCC per year for females from 1999 to 2020.

**Table 1 cancers-16-00606-t001:** Age-standardized incidence rates per 100,000 person-years of keratinocyte skin cancer in the Australian state of Queensland [11], according to the Australian standard population (2001); the United States [12] and Denmark [3], according to the World standard population (2000); and the German federal state of Saarland [3], according to the European standard population (2013).

	Age-Standardized Incidence Rates for Men	Age-Standardized Incidence Rates for Women
Queensland		
2011–2014	3105	2296
United States		
2019	1021	603
Denmark		
2012	126.5	124.8
Saarland		
1970	12.0	9.7
2016	115.6	102.7

**Table 2 cancers-16-00606-t002:** Age-standardized incidence rates per 100,000 person-years of cutaneous squamous cell carcinoma in the Australian state of Queensland [11], according to the Australian standard population (2001); the United States state of Minnesota [18], according to the United States standard population (2010); the United Kingdom [13] and the Netherlands [19], according to the European standard population (2013); Denmark [14], according to the World standard population (2000); and the German federal states of Saarland and Schleswig Holstein [19], according to the European standard population (2013).

	Age-Standardized Incidence Rates for Men	Age-Standardized Incidence Rates for Women
Queensland		
2011–2014	573	371
Minnesota		
2000–2010	207.5	128.8
United Kingdom		
2013–2015	111	42
Netherlands		
1989	35.9	13.1
2021	108	69.7
Denmark		
1978	9.7	4.6
2007	19.1	12.0
Saarland		
1990	26.3	14.9
2019	83.8	35.0
Schleswig Holstein		
1999	39.0	19.6
2020	68.6	36.1

**Table 3 cancers-16-00606-t003:** Age-standardized mortality rates per 100,000 person-years of keratinocyte skin cancer in Australia [21], according to the Australian standard population (2001); the United States [12], according to the World standard population (2000); the Netherlands and the German federal states of Saarland and Schleswig Holstein [19], according to the European standard population (2013).

	Age-Standardized Mortality Rates for Men	Age-Standardized Mortality Rates for Women
Australia		
2016	2.8	1.1
United States		
2019	1.3	0.3
Netherlands		
1989–1991	1.3	0.57
2018–2020	1.21	0.57
Saarland		
1990–1992	1.3	0.56
2017–2019	1.79	0.68
Schleswig Holstein		
1999–2001	0.96	0.33
2016–2018	1.60	0.81

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
