# Peer review of "Epidemiology of Keratinocyte Skin Cancer with a Focus on Cutaneous Squamous Cell Carcinoma"

_cancers, 2024, doi:10.3390/cancers16030606_

Round 1

Reviewer 1 Report

Comments and Suggestions for Authors

I thank the academic editor of Cancers for the possibility of reviewing this interesting manuscript entitled "Epidemiology of Epithelial Skin Cancer with Respect to Cutaneous Squamous Cell Carcinoma" in which the authors report in a detailed and clear manner an epidermal analysis on Nonmelanoma Skin Cancer (NMSC) in Germany, Northern Europe, Australia and the USA. Although at the beginning I wondered what the point of writing a paper on data already well known to the scientific community was, I changed my mind by evaluating the quality of the data presented and the impact they have, seeing and considering that they do not exist in all countries of incidence registers relating to NMSC. The curves presented by the authors are well structured and I have no particular questions and/or observations to make. Recommend to the authors two small additions: 1) add a paragraph relating to the histopathological characteristics of SCC and BCC; 2) carefully check the English language in some sentences.

Comments on the Quality of English Language

moderate changes

Reviewer 2 Report

Comments and Suggestions for Authors

The authors present an interesting and well writen review on epideliomolgy of keratinocyte carcinoma, with special emphasis on SCC. I’d like to add some minor comments:

-          In the abstract, the authors state, "we present the epidemiology of epithelial skin cancer and SCC." This statement is confusing. It would be clearer to specify that they are presenting the epidemiology of epithelial skin cancer, with a focus on SCC, as SCC is included as one entity within the broader category.

-          It is worth noting that some studies already indicate a higher mortality rate for SCC than for Melanoma in the USA. Despite the lower mortality rate for squamous cell carcinoma compared to melanoma, the significantly higher incidences result in absolute numbers surpassing those of melanoma.

-          Since the entire study compares and examines populations from Australia, the USA, and Europe, it would be beneficial to include data from Spanish studies. This inclusion would provide a more comprehensive view of the European continent, not just focusing on Nordic countries with lighter skin phototypes.

Reference:

Tejera-Vaquerizo A, Descalzo-Gallego MA, Otero-Rivas MM, Posada-García C, Rodríguez-Pazos L, Pastushenko I, Marcos-Gragera R, García-Doval I. Skin Cancer Incidence and Mortality in Spain: A Systematic Review and Meta-Analysis. Actas Dermosifiliogr. 2016 May;107(4):318-28. English, Spanish. doi: 10.1016/j.ad.2015.12.008. Epub 2016 Feb 4. PMID: 26852370.

Reviewer 3 Report

Comments and Suggestions for Authors

Suggestions about references: 

Consider reference more recent (Sung H, Ferlay J, Siegel RL, Laversanne M, Soerjomataram I, Jemal A, Bray F. Global Cancer Statistics 2020: GLOBOCAN Estimates of Incidence and Mortality Worldwide for 36 Cancers in 185 Countries. CA Cancer J Clin. 2021 May;71(3):209-249. doi: 10.3322/caac.21660. Epub 2021 Feb 4. PMID: 33538338 ) respect your references 1, 2, 4.

In consideration of references 8-9, please explain if reference 9 is necessary. The reference 8 is more recent than 9 and it reported Incidence trends of nonmelanoma skin cancer in Germany from 1999 explained in reference 9. I suggest remove reference 9.

Reviewer 4 Report

Comments and Suggestions for Authors

The authors reviewed the epidemiology of epithelial skin cancer in Australia, the United States, and the northern Europe. Epithelial skin cancer includes basal cell carcinoma and squamous cell carcinoma and it is the most common neoplasm of white-skinned people. It is nore frequent that malignant melanoma and its onset is largely attributed to the exposure to ultra-violet (UV) radiation. While it seems that an intense UV exposure during childhood and adolescence may induce the development of basal cell carcinoma, squamous cell carcinoma is mainly linked to chronic, cumulative UV damage over decades. 

In the present review paper the authors also emphasized the further perspectives health systems will be confronted with in the next 20 years.

The manuscript is well written and the topic sounds attractive. 

However, I have some points that must be addressed to improve the present manuscript:

1. The main point is that an expanded discussion/introduction about the methods with which skin cancer is diagnosed must be added to the manuscript. The folloqing diagnostic tools must be emphasized : dermoscopy, reflectance confocal microscopy, LC-OCT and histopathology. Please see: PMID: 30357933 and PMID: 35666617.

2. The text will benefit from an English language check.

Comments on the Quality of English Language

Minor editing of english language is required.
